# CLiFT: Compressive Light-Field Tokens for Compute Efficient and Adaptive Neural Rendering

**Zhengqing Wang**[1]    **Yuefan Wu**[1]    **Jiacheng Chen**[1]
**Fuyang Zhang**[1]    **Yasutaka Furukawa**[1,2]

[1]Simon Fraser University        [2]Wayve

## Abstract

This paper proposes a neural rendering approach that represents a scene as "compressed light-field tokens (CLiFTs)", retaining rich appearance and geometric information of a scene. CLiFT enables compute-efficient rendering by compressed tokens, while being capable of changing the number of tokens to represent a scene or render a novel view with one trained network. Concretely, given a set of images, multi-view encoder tokenizes the images with the camera poses. Latent-space K-means selects a reduced set of rays as cluster centroids using the tokens. The multi-view "condenser" compresses the information of all the tokens into the centroid tokens to construct CLiFTs. At test time, given a target view and a compute budget (i.e., the number of CLiFTs), the system collects the specified number of nearby tokens and synthesizes a novel view using a compute-adaptive renderer. Extensive experiments on RealEstate10K and DL3DV datasets quantitatively and qualitatively validate our approach, achieving significant data reduction with comparable rendering quality and the highest overall rendering score, while providing trade-offs of data size, rendering quality, and rendering speed. Check demos and code on the project page: `https://clift-nvs.github.io`.

## 1 Introduction

Global consumption of visual media is skyrocketing, driven by platforms like Instagram, YouTube, and TikTok. Billions of photos and videos are captured, shared, and streamed daily, placing enormous demands on storage and bandwidth. The success of these platforms owes much to advances in visual data compression, allowing high-resolution content to be delivered efficiently across a range of devices and network conditions. From images to high-definition videos, modern compression algorithms have enabled rich visual experiences to become a ubiquitous part of everyday life.

Beyond passive viewing of recorded media, interactive novel view synthesis (NVS) is gaining momentum, allowing users to freely navigate virtual environments. Neural rendering techniques such as Neural Radiance Fields (NeRF) and 3D Gaussian Splatting (3DGS) are the driving forces. Recent research has explored efficiency of neural rendering pipelines, including compression of radiance fields [24, 35], sparse representations [17, 22], and adaptive quality rendering [9, 36]

Along the line of interactive visual media, reconstruction-free novel view synthesis is emerging as a promising direction. Models such as Large View Synthesis Models (LVSM) [12] and Scene Representation Transformers (SRT) [25] synthesize novel views directly, without defining bottleneck geometric and photometric representations by heuristics and reconstructing them. By avoiding explicit reconstruction, these methods would better handle scene dynamics and capture fine-grained visual details directly from the data. Compute-efficient representations and compute-adaptive rendering within these systems would unlock the full potential of NVS technology, driving new applications

39th Conference on Neural Information Processing Systems (NeurIPS 2025).

across real estate (virtual property tours), entertainment (immersive media and games), online shopping (interactive product displays), and autonomous driving (simulation and model validation).

Towards realizing this potential, this paper introduces a new representation and rendering framework centered on the concept of Compressive Light-Field Tokens (CLiFT). CLiFT is a compact set of light field rays with compressed learned embeddings. Concretely, given a set of images with camera poses as input, multi-view encoder tokenizes the images with camera poses. A latent-space K-means algorithm selects a reduced set of rays as cluster centers. Intuitively, the resulting cluster centers preserve coverage due to geometric diversity among rays, while becoming denser in texture-rich regions. Multi-view "condenser" compresses the information of all the embeddings into the centroid tokens to construct CLiFTs At test time, given a target camera pose and a compute budget (i.e., the number of CLiFTs to use), the system collects the corresponding nearby tokens and synthesizes a novel view using a compute-adaptive renderer that is trained to handle a variable number of tokens.

We have evaluated the approach on RealEstate10K [37] and DL3DV [16] datasets, and compared with three state-of-the-art methods, LVSM [12] from the reconstruction-free approach, and MVSplat [3] and DepthSplat [34] from the reconstruction-based approach. Extensive quantitatively and qualitatively validate our approach, achieving significant data reduction with comparable rendering quality and the highest overall rendering score, while providing trade-offs of data size, rendering quality, and rendering speed.

## 2 Related Work

**Light Field Imaging.** Computational light field imaging techniques date back to the 1990s with notable work by Levoy *et al*. on Light Field Rendering [15] and Gortler *et al*. on Lumigraph [10]. Both introduced the idea of capturing a dense array of rays from multiple viewpoints, enabling novel view generation without geometry reconstruction. Subsequent efforts, such as multi-camera arrays [32] and hand-held plenoptic cameras [23], brought these ideas into practical systems. For example, commercial light field cameras like the Lytro (circa 2011) demonstrated post-capture refocusing and viewpoint adjustment. Recent works combine classical light field concepts with neural rendering to enable efficient and accurate novel view synthesis with view-dependent effects [5, 26, 28, 29]. The idea of using light fields (or rays) as a scene representation forms the core of our approach, where we combine this classical concept with modern neural rendering techniques.

**Compressive Sensing.** In the mid-2000s, compressive sensing emerged as a paradigm for capturing signals with far fewer samples than traditional Nyquist sampling theory would require, given the signal is sparse in some basis [1, 4]. In computational imaging, compressive sensing inspired novel camera designs that sample and project measurements into lower-dimensional spaces. Notable examples include the single-pixel camera [6] and coded aperture systems [20] that optically encode high-dimensional scene information into compressed sensor readings. Similar to compressive sensing, our approach selects which rays to store and how to represent them in a compact form for successful view synthesis. A key distinction is that our method learns to compress all the input rays into a compact set of representative rays using neural networks, whereas compressive sensing relies on predefined heuristics to drop information and project to a lower dimension.

**Reconstruction-Based Novel View Synthesis.** Early image-based rendering (IBR) systems relied on reconstructed scene geometry for novel view generation. Photo Tourism is a seminal example, using a planar geometric proxy for rendering—simple yet producing compelling visual experiences [27]. Subsequent work extended this idea by more accurate depth maps to enhance rendering quality [8]. More recently, Neural Radiance Fields (NeRF) introduced a continuous volumetric scene representation as a 5D function that maps spatial location and viewing direction to color and density using an MLP [21]. 3D Gaussian Splatting (3DGS) eliminates neural networks in favor of explicit point-based primitives, representing the scene as millions of 3D Gaussians and rendering images by projecting ("splatting") them with view-dependent shading [13]. These methods typically require per-scene optimization, lack generalization across scenes, and assume dense input coverage. To overcome the limitations, feed-forward Gaussian splatting methods [2, 3, 34] eliminate per-scene optimization by predicting Gaussian parameters in a single forward pass. In parallel, image and video generative models have been incorporated to refine NVS outputs, enhancing realism and temporal consistency [7, 31]. Recent methods like Reconfusion [33] and NerfDiff [11] take a step further by using generative refinement.

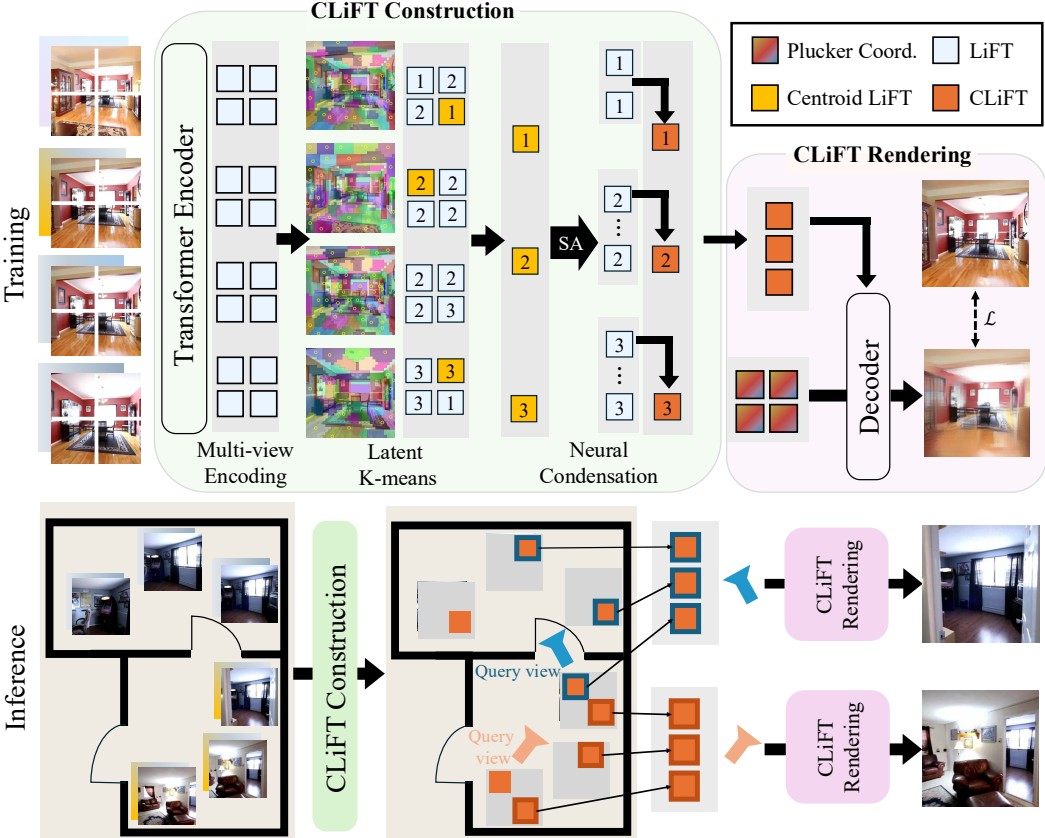

Figure 1: The training and the inference system overview. **Top:** The training consists of three steps: 1) Multi-view encoder, tokenizing the input images; 2) Latent K-means, selecting a representative set of tokens; and 3) Neural condensation, compressing the information of all the tokens into the representative set to produce Compressed Light-Field Tokens (CLiFTs). **Bottom:** At inference time, multi-view images are encoded into CLiFTs following the same process as in training. Given a target view, we collect a relevant set of CLiFTs with simple heuristics and render a novel view.

**Reconstruction-Free Novel View Synthesis.** Reconstruction-free novel view synthesis is gaining attention as an end-to-end solution. The Scene Representation Transformer (SRT) encodes a set of input images into a latent scene representation—a collection of tokens—and renders novel views from this latent in a single feed-forward pass [25]. Another representative work is LVSM (Large View Synthesis Model), a fully transformer-based pipeline that achieves state-of-the-art results from sparse inputs [12]. LVSM includes both an encoder–decoder transformer, which compresses input images into a fixed-length latent code before decoding novel views, and a decoder-only transformer that maps input views directly to output pixels. Multi-view generative models share similar characteristics, producing consistent images across views without reconstructing or generating explicit geometry [14, 18, 19, 30]. This paper advances the frontier of reconstruction-free NVS by introducing Compressive Light-Field Tokens: a compact, variable-size scene representation paired with a renderer that adaptively balances quality and resource usage on demand.

## 3  Compressive Light-Field Tokens (CLiFT)

A light field is a collection of rays, each associated with the radiance information. Compressive light-field tokens (CLiFTs) are a collection of rays, each associated with a latent vector to which a neural encoder compresses the geometry and radiance information of a scene (See Figure 1). Given a target camera pose, a traditional light field rendering synthesizes a color for each target ray, typically by linear interpolation of nearby input rays. CLiFT rendering synthesizes an image (not necessarily per ray) by using nearby CLiFTs via a neural renderer. The framework allows us to control data size,

rendering quality, and rendering speed by two hyper parameters 1) Storage CLiFT count ($N_s$), the number of tokens to represent a scene; and 2) Render CLiFT count ($N_r$), the number of tokens to use in rendering a frame. The section explains the problem, CLiFT construction, and CLiFT rendering.

## 3.1  Problem Definition

This paper tackles the problem of taking $N_c$ images with the camera poses, and constructing $N_s$ CLiFTs as a scene representation. Given a target view, the output is its synthesized image, where $N_r$ CLiFTs tokens are to be selected and used for rendering. The evaluation is based on standard image reconstruction metrics, PSNR, SSIM, and LPIPS. A token (CLiFT) is a $D$ dimensional embedding, associated with a ray. Ray geometries are represented in a global coordinate frame.

## 3.2  CLiFT Construction

Construction of compressed light field tokens (CLiFTs) involves three key steps: encoding multi-view images and camera poses, selecting representative rays, and condensing information into the selected set. The section explains the steps.

**Multi-view Encoding.** Given images with camera poses, we use a standard Transformer encoder to extract our Light Field Tokens (LiFT), which capture both geometry and appearance information. Specifically, for each pixel in each image, we concatenate the 6D Plücker coordinates of the corresponding ray with its normalized 3D color vector. We then patchify non-overlapping 8×8 regions, resulting in $576 = (3 + 6) \times 8 \times 8$ dimensional vectors. [1] Each vector is linearly projected to a token of dimension $D = 768$. This process converts $N_c$ input images of resolution 256×256 into 1024 $\cdot N_c$ tokens. The Transformer encoder performs self-attention across all 1,024 $\cdot N_c$ tokens, producing a total of 4,096 LiFTs for a scene of 4 input views. Specifically, our encoder has six self-attention blocks, each comprising Self-Attention (8 heads), Add & LayerNorm, Feedforward Network, and another Add & LayerNorm. Unlike the encoder-decoder architecture in LVSM, which uses a learnable token to aggregate scene information, our method directly uses the outputs of Transformer encoder, which retains the geometry and appearance of a specific region.

**Latent-space K-means for Ray Selection.** Effective ray selection is crucial, as uniform ray selection across input images leads to two redundancies: 1) Appearance redundancy at texture homogeneous regions; and 2) Geometric redundancy at visual overlaps among different views. Our approach performs latent-space clustering to select representative rays that compactly represent an entire scene. K-means clustering algorithm finds the clusters from LiFT of all the images. The nearest neighbor sample from each center is retained as the cluster centroid. Centroid LiFT will be the storage CLiFTs, and $K$ is set to $N_s$ (i.e, the number of storage CLiFTs).

**Neural Condensation.** A lightweight transformer condenses information from all LiFTs into a set of centroid LiFTs, producing CLiFTs. Self-attention operates over the centroid LiFTs to enable information exchange across clusters. Within each cluster, cross-attention uses the centroid LiFT as the query and the remaining LiFTs as keys and values. To preserve the pretrained latent space, a zero-initialized linear layer aggregates features back into the centroid LiFTs.

Concretely, the condensation network consists of two transformer decoder blocks. Each block includes inter-cluster self-attention (8 heads), intra-cluster cross-attention (8 heads), and standard Add & LayerNorm operations. Let $T_k \in \mathbb{R}^D$ denote the embedding of the $k$-th centroid LiFT, and $\{T_{k,i} \mid i\}$ represent the remaining LiFTs assigned to the $k$-th cluster. SA($\cdot$), CA($\cdot, \cdot$), and FFN($\cdot$) denote the self-attention, cross-attention, and feed-forward network layers, respectively. $W_z$ is a zero-initialized linear projection operator $W_z \in \mathbb{R}^{D \times D}$. Each decoder block updates the centroid

---

[1]As explained in §4, we use RealEstate10K [37] and DL3DV [16] datasets. Image resolutions are 256×256 and 256×448, respectively. The section uses RealEstate10K as an example to explain the feature dimensions. The patch size is the same for DL3DV, resulting in 1,792 $\cdot N_c$ tokens instead.

embeddings as follows, producing CLiFTs after the second block:

$$\{\hat{T}_k\} \leftarrow \text{SA}\left(\{\text{LN}(T_1), \text{LN}(T_2), \cdots\}\right), \tag{1}$$

$$\forall k \quad \hat{T}_k \leftarrow \text{CA}(\hat{T}_k, \{\text{LN}(T_{k,1}), \text{LN}(T_{k,2}), \cdots\}), \tag{2}$$

$$\forall k \quad \hat{T}_k \leftarrow \text{FFN}(\text{LN}(\hat{T}_k)), \tag{3}$$

$$\forall k \quad T_k \leftarrow T_k + W_z(\hat{T}_k). \tag{4}$$

### 3.3 CLiFT Rendering

CLiFTs enable a flexible and efficient rendering mechanism. Given a target view and the render CLiFT count (i.e., the number of tokens to use), we collect a set of relevant CLiFTs, which are then used by a neural renderer to synthesize the view.

**Neural Renderer.** The renderer is a simple Transformer decoder where the target view serves as the query and the selected CLiFTs serve as keys and values. Specifically, we initialize a 2D grid of Plücker coordinates at each pixel and patchify them. Each patch is represented as a 384-dimensional vector derived from the $6\times8\times8$ Plücker coordinates, which is projected to 768 dimensions via a linear layer. The rendering network has six decoder blocks, each comprising Self-Attention (among query tokens), Add & LayerNorm, Cross-Attention (from CLiFTs to query tokens), Add & LayerNorm, Feedforward Network, and Add & LayerNorm. The number of heads is eight in SA and CA layers. The decoder output is passed through a linear projection, followed by a sigmoid activation to map it to the RGB space, and then an unpatchify operation to reconstruct the full-resolution image. We use a combination of L2 loss and perceptual loss (with a 0.5 weight) following LVSM [12], applied to the normalized RGB intensities. During training, we randomly vary the number of CLiFTs passed to the decoder, allowing it to learn how to handle different token counts and enabling dynamic trade-offs between rendering quality and computational cost.

**Token Selection.** The token selection algorithm employs a simple heuristic. Let $N_r$ denote the number of CLiFTs used for rendering. To ensure spatial coverage of the target view, we divide the view into a $16\times16$ grid of patches and pad it with a 4-patch margin on all sides, resulting in a $24\times24$ grid. For each patch, we cast a ray through its center and retrieve the $N_r/(24 \times 24)$ closest CLiFTs from a pool of $N_s$ storage CLiFTs. The distance between a patch and a CLiFT is computed using a heuristic based on their associated rays (ray origins and directions); details are provided in the supplementary material. Since CLiFTs may be selected multiple times, we greedily accumulate CLiFTs in order of their minimum distance to any patch ray until $N_r$ unique tokens are selected.

## 4 Experiments

### 4.1 Experimental Setup

**Datasets.** We use two scene datasets RealEstate10K [37] and DL3DV [16], following the recent literature [3, 12, 34]. [2] We preprocess videos and create training/testing video clips in exactly the same as PixelSplat [2] for RealEstate10K and DepthSplat [34] for DL3DV. The only difference is the number of images to use from each clip for training and testing. Concretely, LVSM [12], MVSplat [3], and DepthSplat [34] used 2 images for training and testing for RealEstate10K. MVSplat and DepthSplat used 2-6 images for training and testing for DL3DV. We use 4-6 images for training and 4-8 images for testing in both datasets to handle larger scenes.

**Evaluation Metrics.** We evaluate the rendered image quality by PSNR, LPIPS, and SSIM. To assess computational efficiency, we report data size, rendering FPS, and rendering FLOPs, where FLOPs are theoretical numbers instead of measured ones.

**Training Details.** We adopt a two-stage training strategy. The first stage is to train the multi-view encoder where the neural renderer is directly connected to back-propagate gradients without the latent K-means or the condensation modules. In this case, all the tokens from the encoder are passed to the renderer through cross-attention. The second stage uses all the modules while freezing the multi-view encoder. Since K-means is not efficient for online sampling, we pre-compute the cluster assignments

---

[2]RealEstate10K is under Creative Commons Attribution 4.0 International License. DL3DV is under DL3DV-10K Term of use and Creative Commons Attribution-NonCommercial 4.0 International License.

offline and use them in the second stage. For experiments on the 256×256 RealEstate10K, the first and the second stages train for 90,000 steps with a batch-size 64 and 50,000 steps with a batch-size 80, respectively. For DL3DV (256×448), we finetune the RealEstate10K-pretrained model for 100,000 steps with a batch size of 24 in the first stage and 32 in the second stage. Both datasets use the same cosine learning rate scheduler with a 2500-step warmup. The peak learning rate is $4 \times 10^{-4}$ for RealEstate10K and $2 \times 10^{-4}$ for DL3DV, with the learning rate of the renderer scaled by 0.1 in the second stage. We use four NVIDIA RTX A100 GPUs to train our model. Training takes approximately 3 days on RealEstate10K and 5 days on DL3DV.

**Baselines.** We compare with three state-of-the-art methods, one from the reconstruction-free approach and the other two from the reconstruction-based approach.

• *LVSM* [12] is a state-of-the-art reconstruction-free method. While a publicly available checkpoint exists for RealEstate10K, it corresponds to a significantly larger model, featuring twice as many blocks in both the encoder and decoder, and was trained using 64 A100 GPUs with only 2 input views. To enable a fair comparison, we trained the LVSM-ED system under our setting: using 6 blocks in both the encoder and decoder, and 4 input views. Given the substantial compute requirements of LVSM (i.e., 64 A100 GPUs as reported by the authors), we chose to compare on RealEstate10K. As the LVSM code was not available during our initial experiments, we first reproduced their system independently, and later refined our implementation based on the official code after its release.

• *MVSplat* [3] and *DepthSplat* [34] are state-of-the-art feedforward splatting-based novel view synthesis systems. They achieve strong performance on benchmarks like RealEstate10K and DL3DV by leveraging explicit 3D scene representations and depth-guided 3D Gaussian splatting for rendering. We use publicly available checkpoints for MVSplat and DepthSplat (i.e., 2-view case for RealEstate10K and 2, 4, or 6-view cases for DL3DV).

## 4.2 Main Results

Figure 2 and Figure 4 show the main evaluation results on the RealEstate10K and DL3DV datasets. In the plots, the x-axis represents the data size of the scene representation: Storage CLiFTs for our method, decoder input tokens for LVSM, and splats for MVSplat and DepthSplat. The y-axis reports PSNR, SSIM and LPIPS.

None of the baselines support controlling the data size and require a separately trained model for each data point. Many baseline points are missing from the plots because we either use publicly available checkpoints (for MVSplat and DepthSplat) or train a new model (§4.1). In contrast, our method trains a single model per dataset and supports fine-grained control over both the storage data size and the render data size via the numbers of Storage ($N_s$) and Render ($N_r$) CLiFT tokens.

The plots show that CLiFT achieves comparable PSNR with approximately 5–7× less data size than MVSplat and DepthSplat, and about 1.8× less than LVSM, highlighting the effectiveness of our compressed tokens and overall scene representation. CLiFT also attains the highest overall PSNR with significantly lower data usage. Qualitative results in Figure 4 support these findings: our method preserves sharp appearance details that are closer to the ground truth and maintains high visual fidelity even under strong compression, with only minor loss in high-frequency content.

## 4.3 Ablation Studies

**CLiFT Construction.** To demonstrate the effectiveness of our individual components, Figure 3 compares three variants of our system: 1) The full system (blue); 2) The system without the condenser (green); and 3) The system without both the condenser and latent K-means (red), where tokens are selected randomly instead. We evaluate all variants on 92 randomly selected scenes to ensure consistent and fair comparisons, using an NVIDIA RTX A6000 GPU.

Figure 3 shows how rendering quality (PSNR, LPIPS, SSIM), rendering speed (FPS), and rendering cost (FLOPs: theoretical number instead of measured) change with different compression rates. Specifically, we fix the number of CLiFT tokens for storage ($N_s$) and rendering ($N_r$) to be the same, and vary them together. When the compression rate is low (<2×), all variants perform similarly since even randomly selected tokens provide reasonable coverage. However, at high compression rates, the gap between random selection and K-means selection becomes significant: random selection fails to capture representative tokens, while K-means selects tokens that better summarize the scene. Figure 5 visualizes the selected centroids and their associated cluster members. The clustering is performed

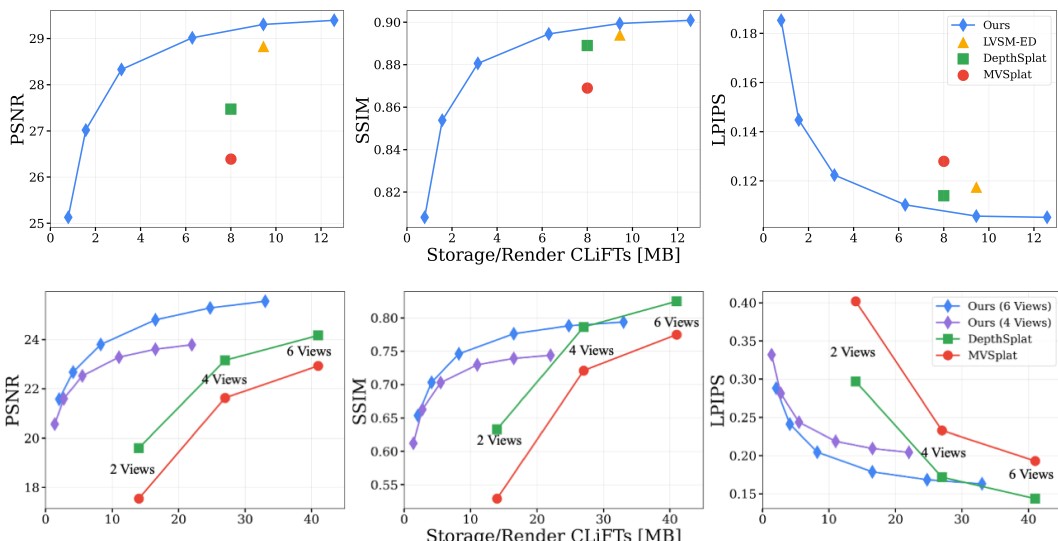

Figure 2: Main evaluation results on the RealEstate10K dataset (top) and the DL3DV dataset (bottom), comparing our approach with three baseline methods (LVSM-ED [12], DepthSplat [34], and MVSplat [3]). The x-axis is the data size of the scene representation, and the y-axis is the rendering quality (PSNR, SSIM and LPIPS). Our approach CLiFT can flexibly change the data size (i.e., number of tokens) with one trained model, achieving significant data size reduction with comparable rendering quality and the highest overall PSNR, while providing trade-offs among data size, rendering quality, and rendering speed.

Table 2: We fix the number of storage CLiFTs to represent a scene ($N_s$=4096), then vary the number of render CLiFTS (how many tokens to use for rendering) on-the-fly and measure rendering quality (PSNR), rendering speed (PSNR), and rendering cost (theoretical number as GFLOPs).

| Metrics | Render CLiFTs | | | | |
|---|---|---|---|---|---|
| | 4096 | 3072 | 2048 | 1024 | 512 |
| PSNR | 26.72 | 26.71 (-0.01) | 26.56 (-0.16) | 25.75 (-0.97) | 23.89 (-2.83) |
| GFLOPs | 70.6 | 63.3 (-10%) | 56.1 (-21%) | 48.9 (-31%) | 45.23 (-36%) |
| FPS | 54.3 | 53.89 (-1%) | 66.44 (+22%) | 80.77 (+49%) | 90.15 (+66%) |

jointly across multiple views, encouraging tokens from different images to form non-redundant clusters. Furthermore, clusters tend to grow larger in texture-homogeneous regions.

**Effectiveness of K-means clustering.** We also evaluate the effectiveness of latent K-means clustering by comparing it against a simple patch-wise grouping baseline. The baseline divides the image into non-overlapping local patches, assigning tokens uniformly across regions, whereas latent K-means adaptively clusters tokens in latent space. As shown in Table 1, latent K-means consistently outperforms patch-wise clustering across both token budgets, with especially large gains under stronger compression (e.g., +2.24 PSNR at 256 tokens with condensation). This improvement arises

Table 1: Ablation study comparing latent K-means clustering with a simple patch-wise grouping baseline. We report PSNR under two token budget settings (256 and 1024), with and without the condenser.

| Method | 256 tokens | 1024 tokens |
|---|---|---|
| Patch-wise | 22.64 | 27.48 |
| Patch-wise + Condenser | 22.97 | 27.55 |
| K-means | 24.46 | 28.17 |
| K-means + Condenser | **25.21** | **28.41** |

because patch-wise clustering distributes capacity evenly, while latent K-means allocates more tokens to informative regions. As illustrated in Figure 11 and Figure 12, this adaptive allocation leads to better visual quality when budgets are tight.

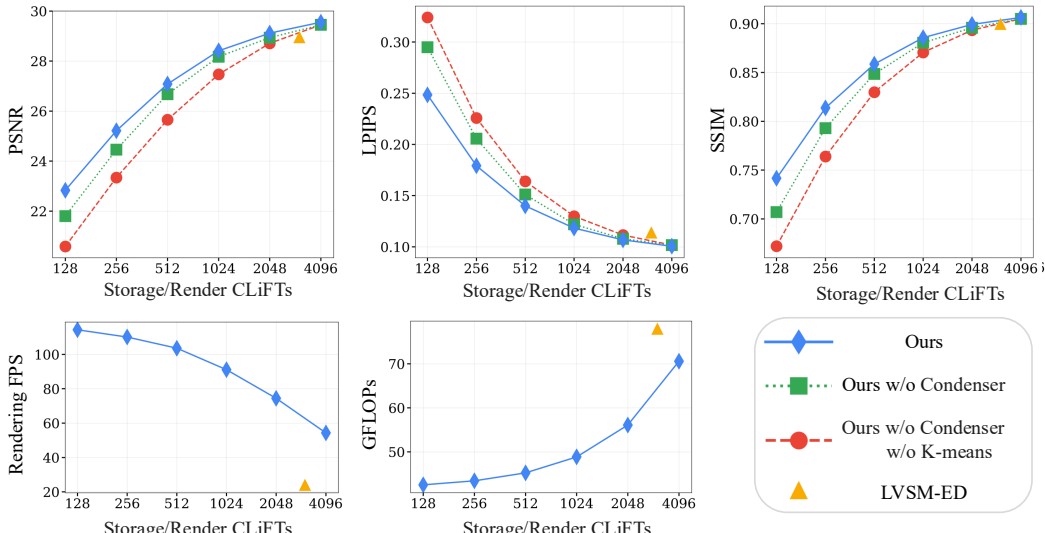

Figure 3: Ablation studies on our individual components, in particular, latent K-means and neural condensation. The plots compare three variants of our system by dropping latent K-means and neural condensation one by one from the system, while varying the data size. Specifically, the x-axis is the size of the scene representation. The y-axis is rendering quality (PSNR, LPIPS, and SSIM), rendering speed (FPS), or rendering cost (FLOPs), measured on an NVIDIA RTX A6000 GPU.

**CLiFT Rendering.** A CLiFT token is associated with a specific ray, encoding localized geometry and appearance. This ray-based design enables intuitive and efficient token selection heuristics at inference time (§3.3), allowing the renderer to control the trade-off between quality and speed on the fly. For example, in large scenes with multiple rooms, rendering a view within one room can be done without using tokens from unrelated areas. In contrast, methods like LVSM represent the entire scene using a fixed set of global tokens, which limits their ability to selectively render parts of the scene or adjust computation dynamically.

We validate this ability on large scenes from the RealEstate10K dataset. Specifically, we construct a large-scene test set by filtering for scenes with more than 200 frames, as higher frame counts typically indicate broader camera motion. We use 50 such scenes for evaluation, with 8 input views per scene to ensure sufficient coverage. As shown in Table 2, our method supports on-the-fly adjustment of rendering cost by varying the number of tokens used, achieving lower FLOPs and higher FPS when needed, or allocating more tokens for higher quality.

## 5 Conclusions and Future Challenges

We introduced CLiFT, a compressive light-field token representation paired with a compute-adaptive transformer renderer. Experiments on RealEstate10K and DL3DV demonstrate that CLiFT achieves higher data reduction rate with comparable rendering quality and the highest overall rendering score, while capable of controlling the trade-offs between data size, rendering quality, and rendering speed. Our current system exhibits two typical failure modes. The first occurs when camera motions deviate significantly from the training distribution. As shown on the left of Figure 6, RealEstate10K training data primarily consists of smooth translations with minor rotations, making it difficult to generalize to more complex motions. The second failure mode arises in large scenes where target views are not adequately covered by the input views, resulting in blurry renderings. This is illustrated on the right of Figure 6 with examples from DL3DV. A promising future direction is to extend our framework by incorporating generative priors to improve rendering quality in unseen or occluded areas.

**Broader Impacts.** On the positive side, our method has potential applications in digital media content consumption, enabling more efficient and flexible rendering for immersive experiences. However, like generative models, it raises concerns around misuse—particularly in the creation of deep-fakes. We emphasize the importance of responsible deployment and clear content provenance.

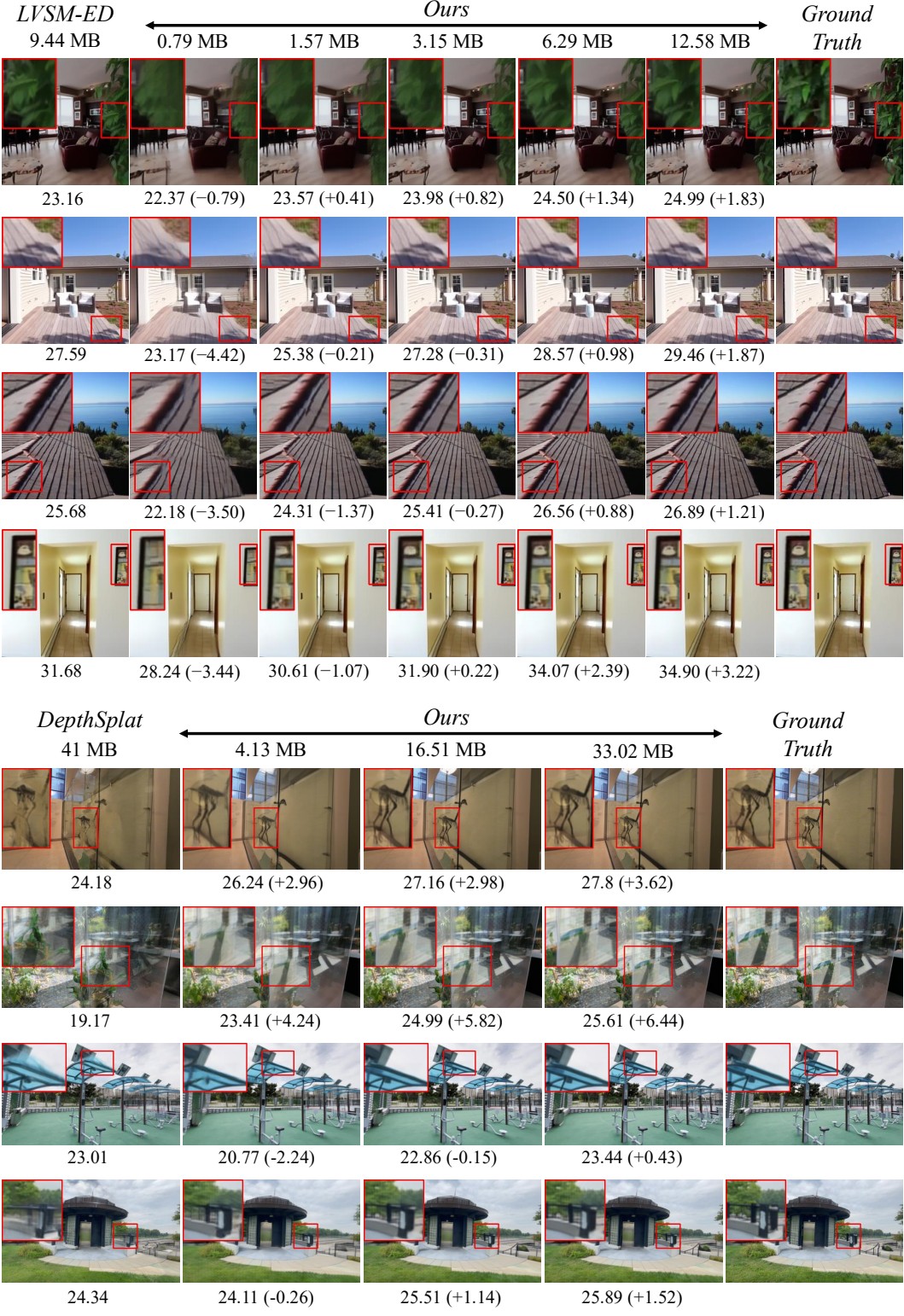

Figure 4: Qualitative rendering results of the baselines and ours with different data size (i.e., the number of CLiFT tokens for ours). **Top**: Ours vs. LVSM [12] on RealEstate10K. **Bottom**: Ours vs. DepthSplat [34] on DL3DV. The PSNR value is recorded under each rendering.

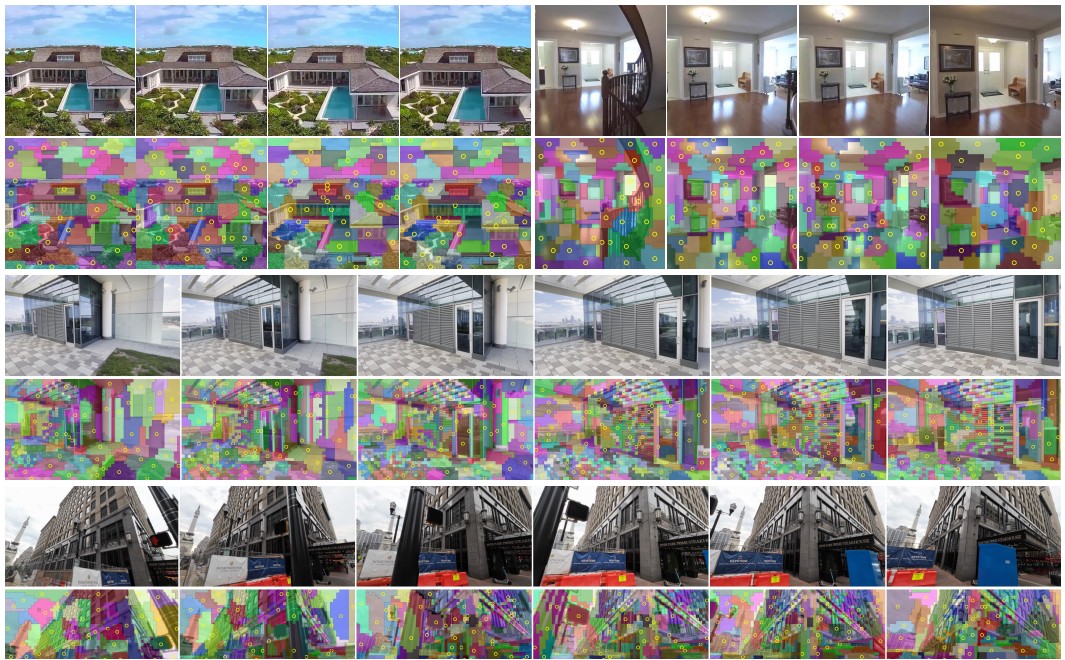

Figure 5: Visualization of the latent K-means clustering, where $K=N_s=128$. Each color represents a cluster, and the yellow ring indicates the centroid token. Note that clustering is performed across multiple views, so a single cluster can span multiple images. As a result, some clusters may not have a visible centroid in a given image.

RealEstate10K                                      DL3DV

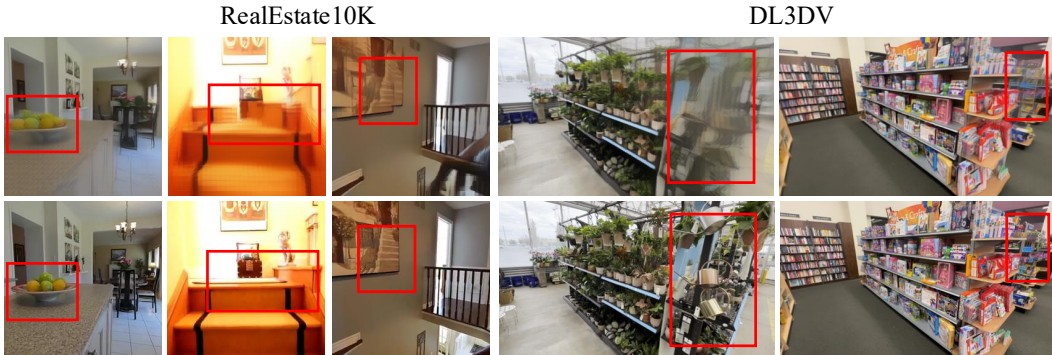

Figure 6: Typical failure cases are (Left) Camera motions deviating too much from the training distributions in RealEstate10K; and (Right) Target views not covered by the input images in DL3DV. The top row shows our results, and the bottom row shows the ground truth.

## Acknowledgments

We thank Haian Jin for helpful discussions on reproducing LVSM and training on the DL3DV dataset. We thank Jiaqi Tan for assistance with the project page and demo design. This research is partially supported by NSERC Discovery Grants, NSERC Alliance Grants, and John R. Evans Leaders Fund (JELF). We thank the Digital Research Alliance of Canada and BC DRI Group for providing computational resources.

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

# Appendix

The appendix provides:

⬦ §A: Additional more rendering results and intermediate visualizations (i.e., clusteirng and token-selection results).

⬦ §B: Implementation details of the latent K-means algorithm (during CLiFT construction) and the token selection algorithm (during CLiFT rendering).

Please check our project page at `https://clift-nvs.github.io`, which shows rendering results at different compression rates and compares with the baseline methods.

## A    Additional Rendered Images

Figure 4 of the main paper presents rendering results under varying data size constraints. We provide additional qualitative examples on RealEstate10K (see Figure 7 and Figure 8) and DL3DV (Figure 9 and Figure 10). These examples cover a finer range of compression rates, extending up to 32×, and highlight the robustness of our method across varying levels of compression. We also show more K-means clustering results in Figure 11 and Figure 12.

## B    Additional implementation details

### B.1    Details of K-means algorithm

During training, we precompute cluster assignments after the multi-view encoder training and before the condensation training, using *faiss.Kmeans* which supports GPU acceleration. At test time, we use *sklearn.cluster.KMeans* for better accuracy.

### B.2    Details of Token Selection

Rendering a specific region within a large scene does not require all tokens. Using only the tokens essential to the target view improves FPS and reduces FLOPs, with minimal impact on PSNR. Algorithm 1 is our token selection algorithm.

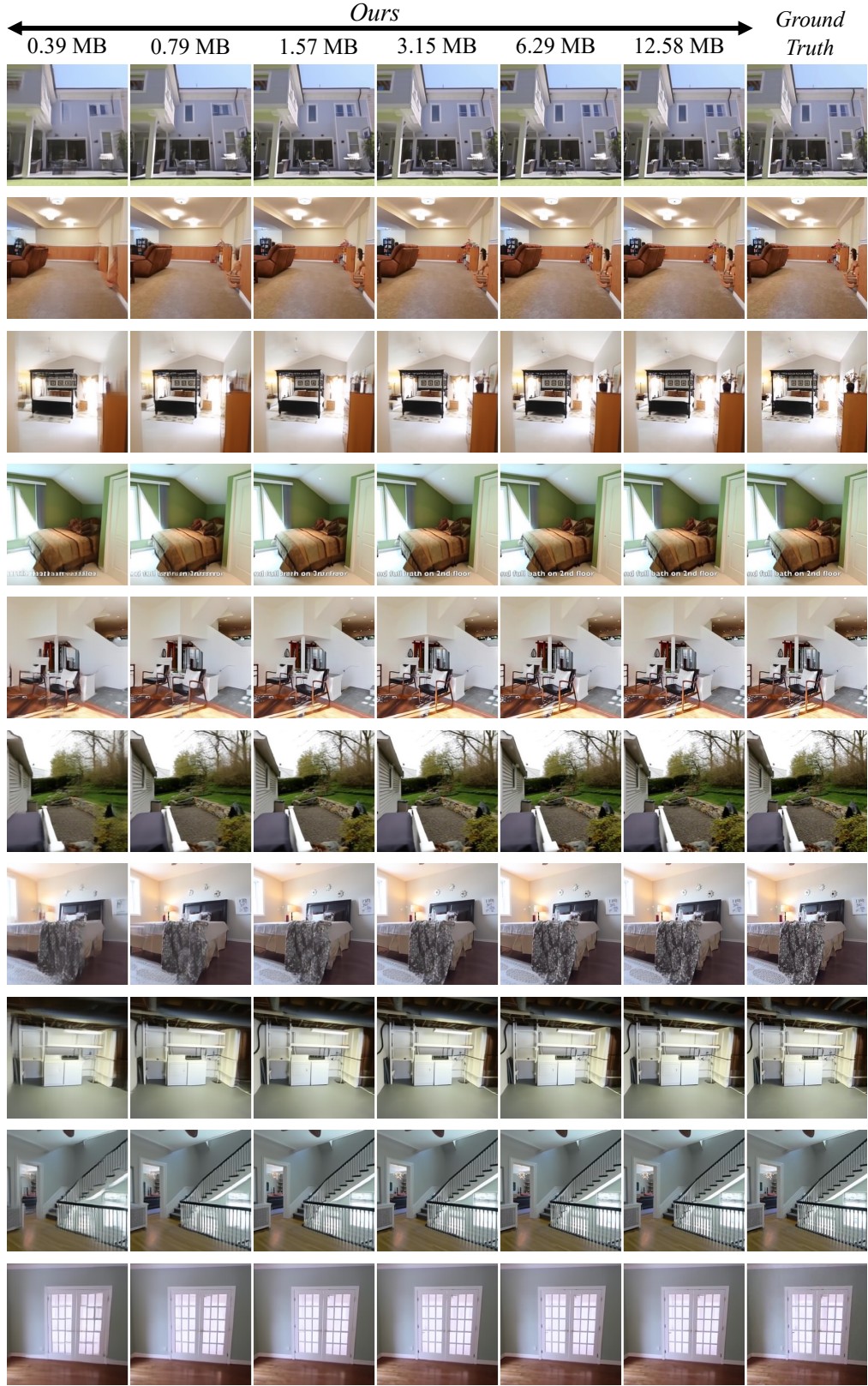

Figure 7: Additional qualitative results on the RealEstate10K dataset.

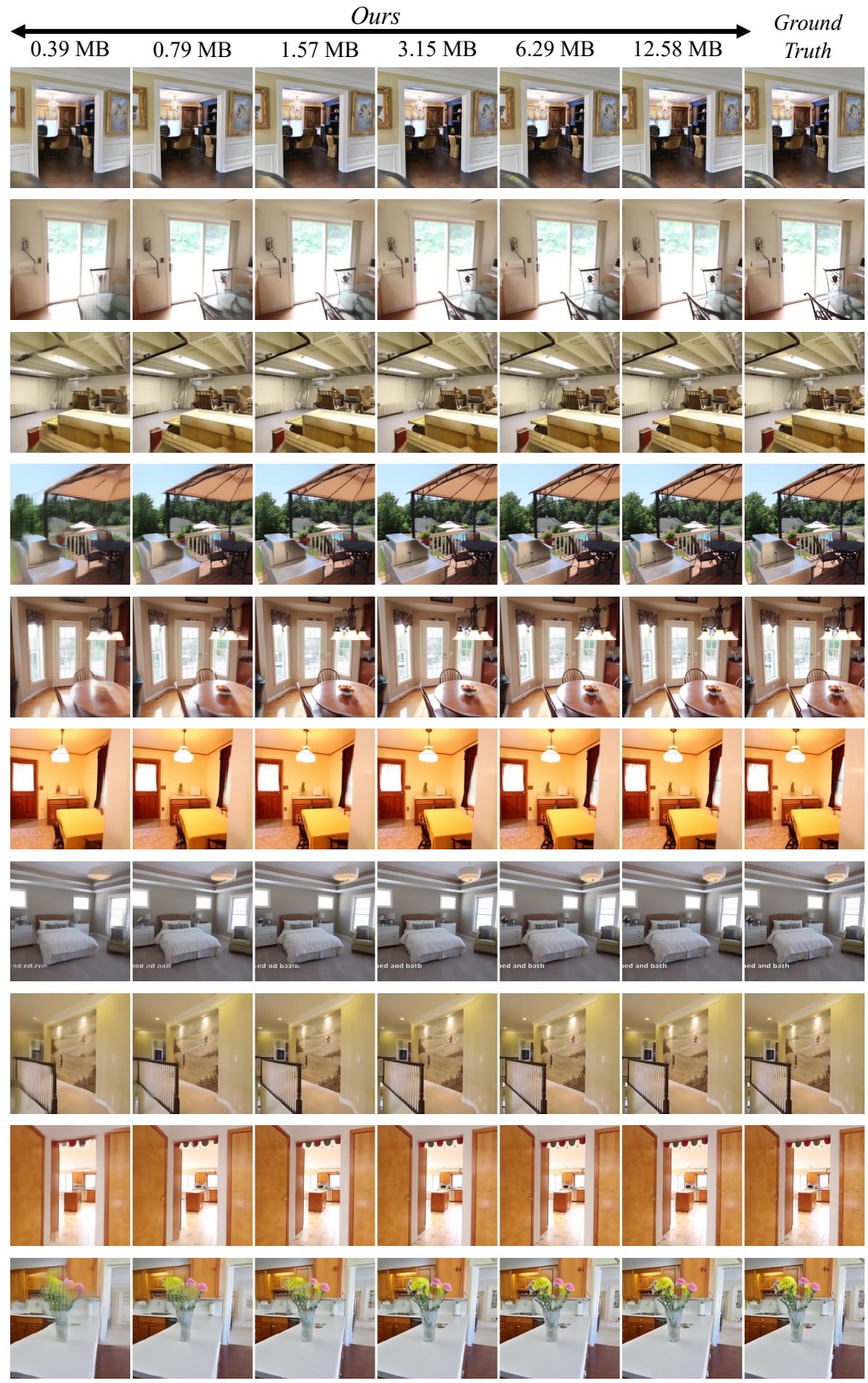

Figure 8: Additional qualitative results on the RealEstate10K dataset.

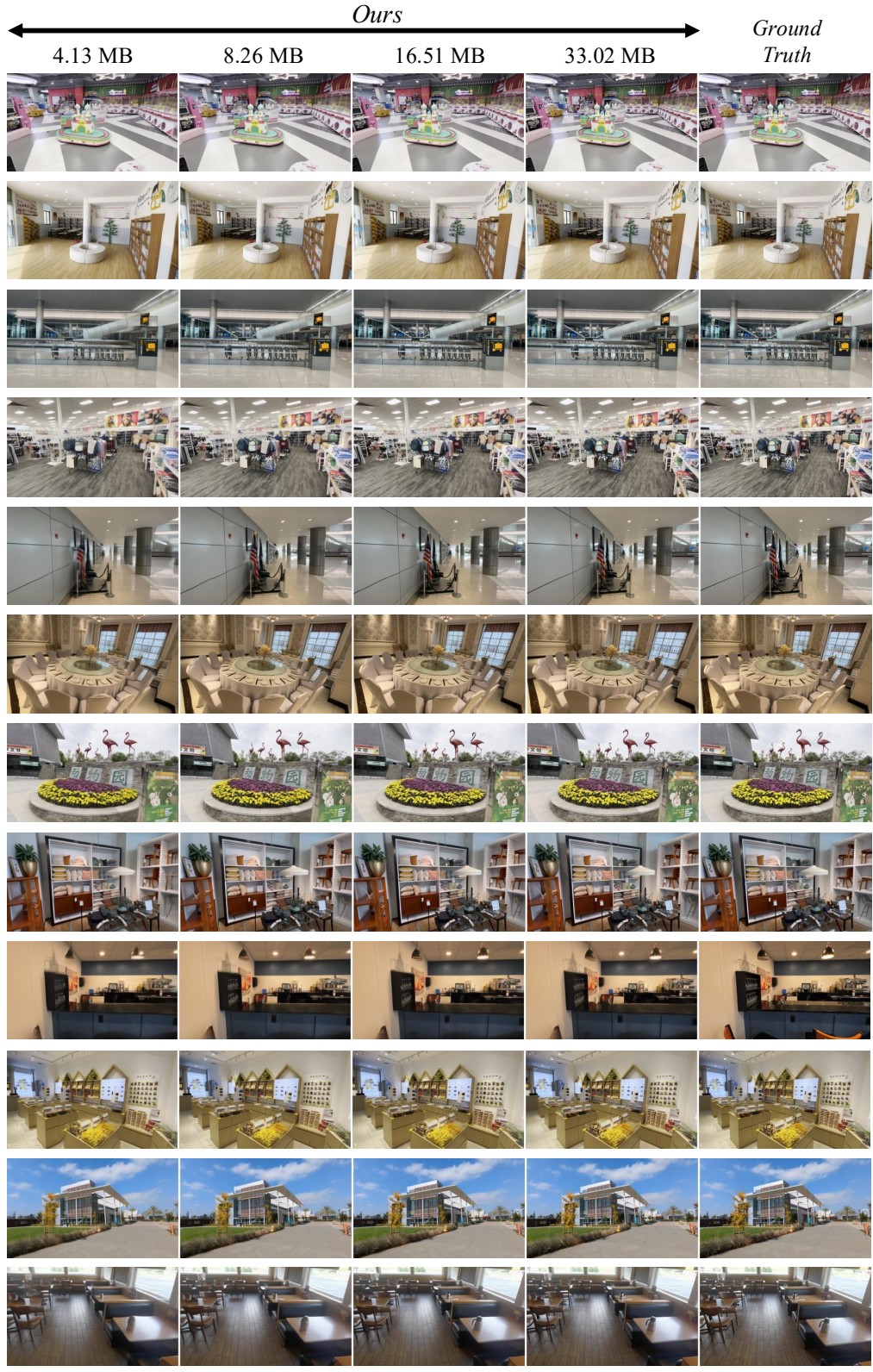

Figure 9: Additional qualitative results on the DL3DV dataset.

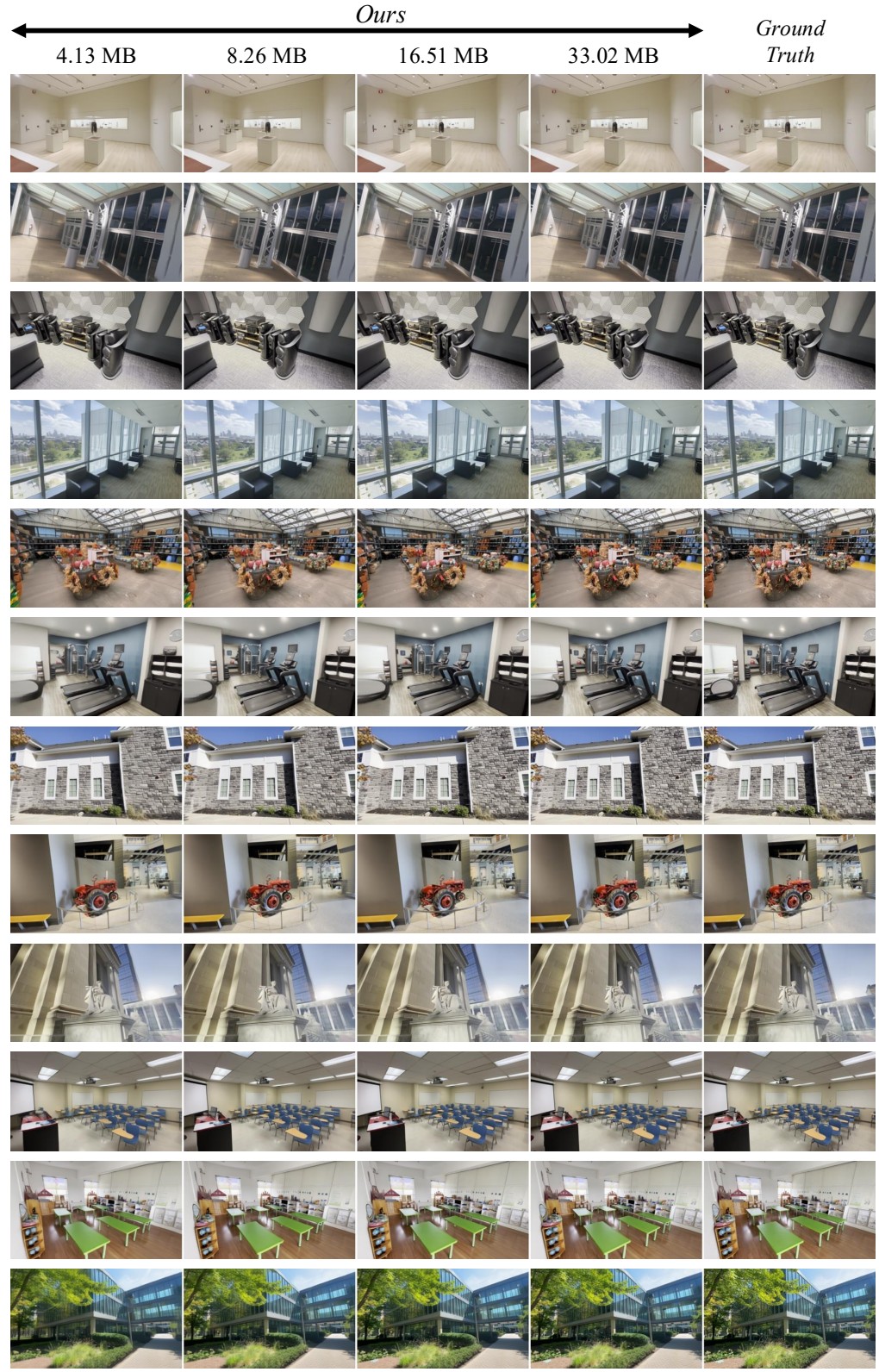

Figure 10: Additional qualitative results on the DL3DV dataset.

---

**Algorithm 1** Token Selection Algorithm

---

1: **Definitions:**
2:     $\theta$: Ray-angle distance between patches
3:     $\delta$: Camera center distance, $\delta = \|o_t - o_k\|$
4:     $m_k$: Last frame token mask
5:     $P$: Number of rays per patch (e.g., $P = 16 \times 16$)
6:     $N$: number of condition images
7:     $w_{\text{angle}} = 1.0$, $w_{\text{dist}} = 0.02$, $w_{\text{mask}} = -0.03$
8:     momentum factor $\eta = 0.5$
9: **Input:**
10:     Target view ray $(o_t, d_t)$ with $o_t \in \mathbb{R}^3$, $d_t \in \mathbb{R}^{P \times 3}$;
11:     Condition view rays $\{(o^k, d^k)\}_{k=1}^N$ with $o^k \in \mathbb{R}^3$, $d^k \in \mathbb{R}^{P \times 3}$;
12:     Last frame token mask $\{m^k\}_{k=1}^N$
13:     Total number of tokens to select $T$
14: **Output:** Selected token indices $\mathcal{I}$
15: Downsample target and condition view rays into $16 \times 16$ patches
16: Expand each target patch to a $24 \times 24$ region, including out-of-image rays, to incorporate edge-adjacent context and avoid boundary under-coverage
17: Set number condition rays per target patch $n \leftarrow T \, / \, / \, (24 \times 24)$
18: Initialize selected patches $\mathcal{I}_{\text{patch}} \leftarrow \emptyset$
19: **// Patch-wise selection**
20: **for** each patch $P_i$ in expanded target patches **do**
21:     **for** each condition view $k$ from 1 to $N$ **do**
22:         **for** each condition patch $\bar{P}^{k,j}$ in condition view **do**
23:             Compute ray-angle distance $\theta$ between rays in $P_i$ and $\bar{P}^{k,j}$
24:             Compute camera center distance between target and condition view $\delta = \|o_t - o^k\|$
25:             Retrieve last frame mask $m^k$
26:             Compute frame distance: $D_k^{i,j} = w_{\text{angle}} \cdot \theta + w_{\text{dist}} \cdot \delta + w_{\text{mask}} \cdot m^k$
27:             Apply momentum: $D_i^{k,j} \leftarrow (1 - \eta) \cdot D_i^{k,j} + \eta \cdot \hat{D}_i^{k,j}$
28:             Store previous step objective: $\hat{D}_i^{k,j} \leftarrow D_i^{k,j}$
29:         **end for**
30:     **end for**
31:     Let $\mathcal{D}_i = \{D_i^{k,j} \mid \forall k, j\}$
32:     $\mathcal{I}_{\text{local}}^i \leftarrow \text{Top}_n\left(\text{argsort}_{\text{asc}}(\mathcal{D}_i)\right)$
        $\mathcal{I}_{\text{patch}} \leftarrow \mathcal{I}_{\text{patch}} \cup \mathcal{I}_{\text{local}}^i$
33: **end for**
34: Compute unique set: $\mathcal{I}_{\text{unipatch}} \leftarrow \text{unique}(\mathcal{I}_{\text{patch}})$
35: Let $T_{\text{rest}} \leftarrow T - |\mathcal{I}_{\text{unipatch}}|$
36: **// Global fallback selection**
37: Initialize set of fallback distances: $\mathcal{D}_{\text{global}} \leftarrow \emptyset$
38: **for** each condition view $k$ from 1 to $N$ **do**
39:     **for** each condition patch $j$ in view $k$ **do**
40:         Compute $\tilde{D}^{k,j} = \min_i D_i^{k,j}$                              // best match to any target patch
41:         $\mathcal{D}_{\text{global}} \leftarrow \mathcal{D}_{\text{global}} \cup \{\tilde{D}^{k,j}\}$
42:     **end for**
43: **end for**
44: $\mathcal{I}_{\text{global}} \leftarrow \text{Top}_n\left(\text{argsort}_{\text{asc}}(\mathcal{D}_{\text{global}})\right)$
45: **return** $\mathcal{I} = \mathcal{I}_{\text{patch}} \cup \mathcal{I}_{\text{global}}$

---

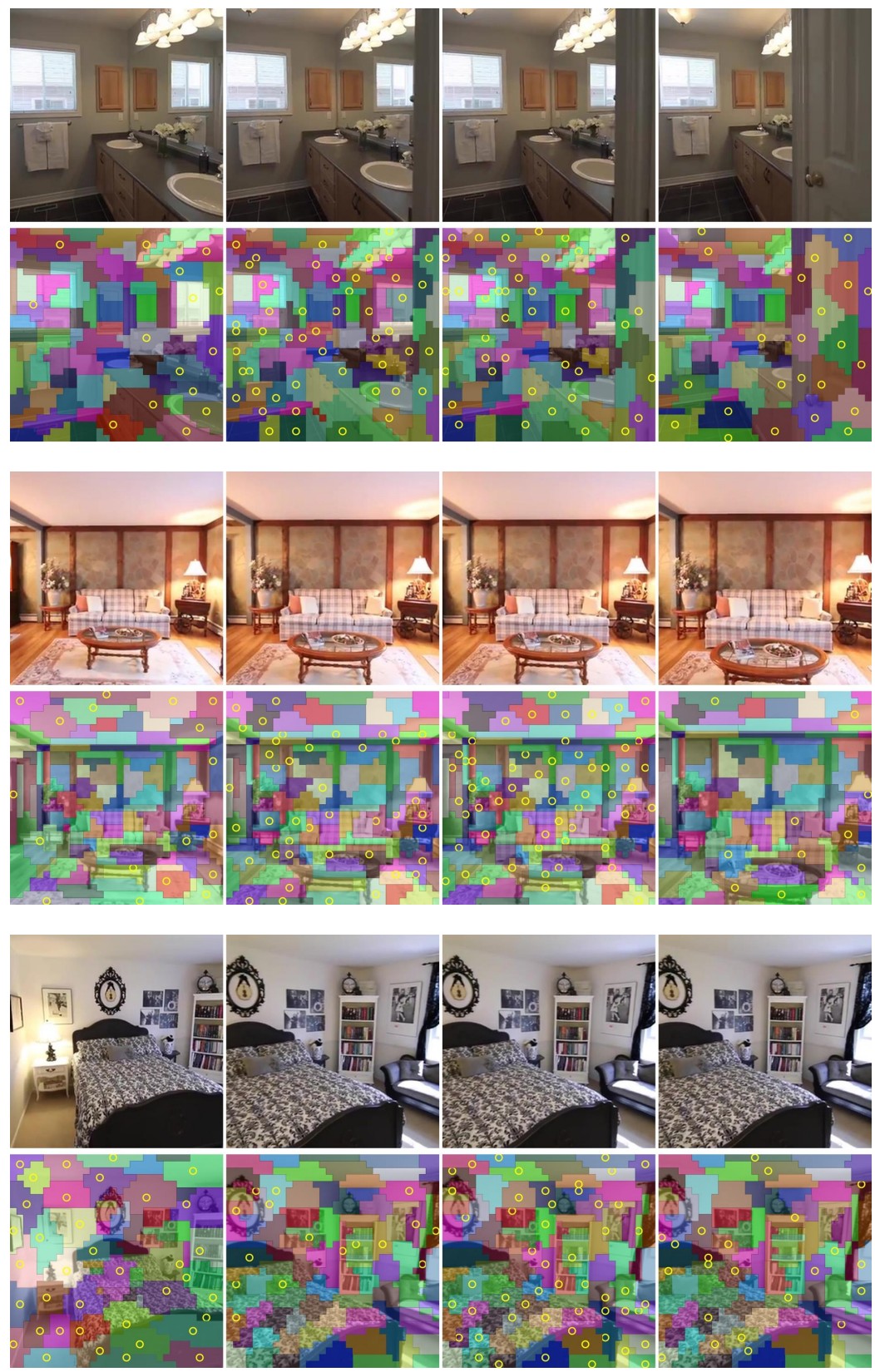

Figure 11: Additional visualization of the latent K-means algorithm for the RealEstate10k dataset.

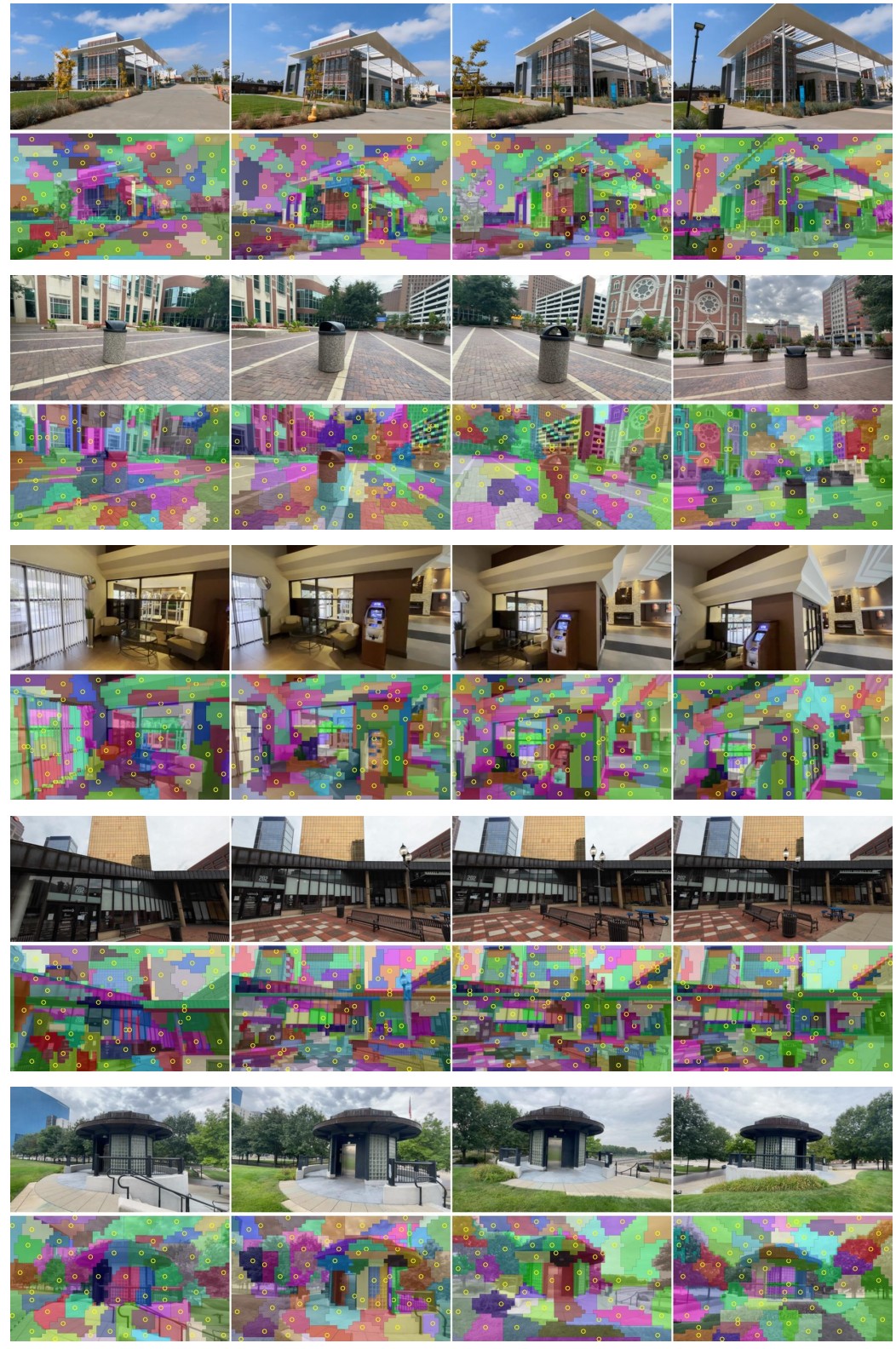

Figure 12: Additional visualization of the latent K-means algorithm for the DL3DV dataset.

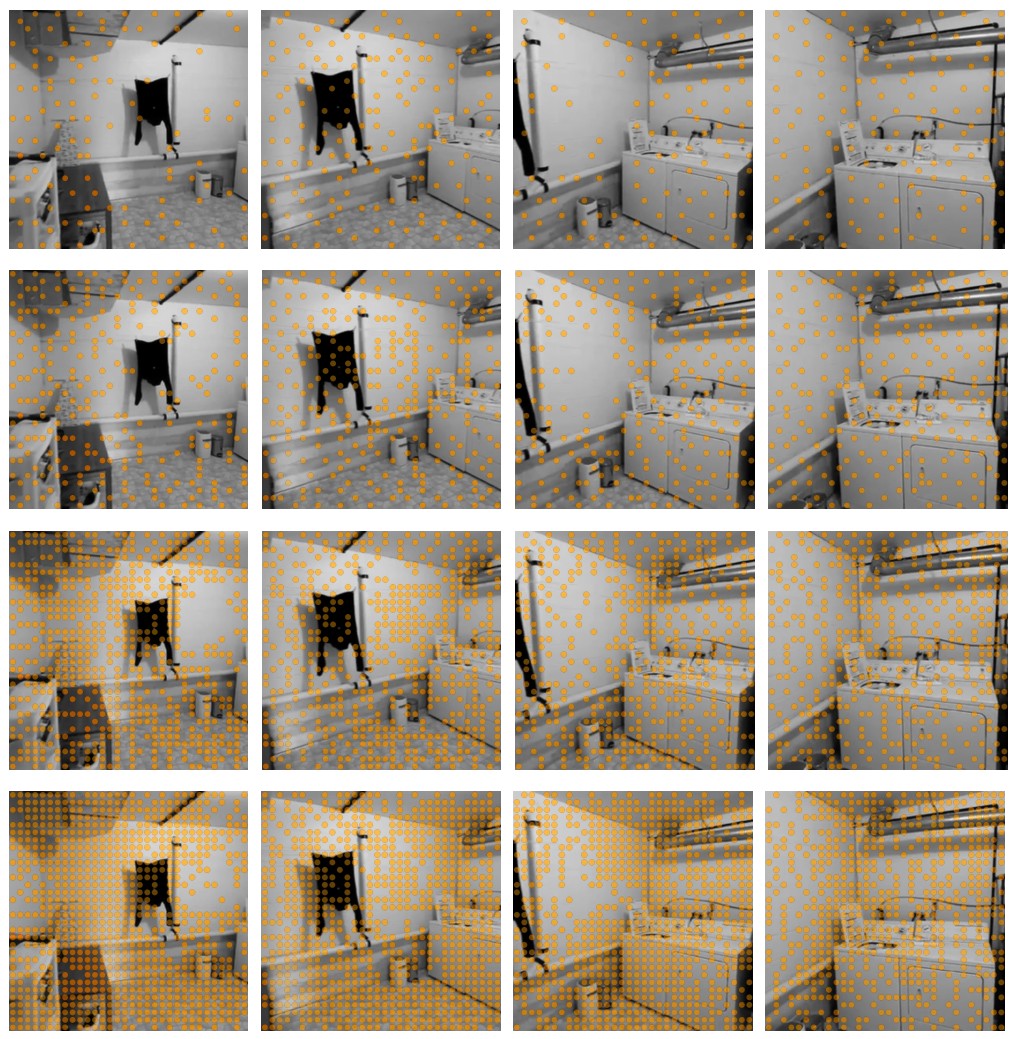

Figure 13: Additional visualization of the latent K-means algorithm with different values of K, which is 512, 1024, 2048, and 3072 from the top.

