# OpenReview forum: "CLiFT: Compressive Light-Field Tokens for Compute Efficient and Adaptive Neural Rendering"
_NeurIPS.cc/2025/Conference — NeurIPS 2025 spotlight_

### Official Review · Reviewer_6dxz · 2025-06-27

**Clarity:** 3
**Significance:** 2
**Originality:** 3
**Rating:** 5
**Confidence:** 4

**Summary:**

This paper introduces CLiFT (Compressive Light-Field Tokens), a new token-based scene representation for compute-efficient and adaptive neural rendering. The method targets explicit reconstruction-free novel view synthesis (NVS), enabling a single trained model to produce high-quality renderings while flexibly trading off between data size, rendering speed, and visual fidelity. CLiFTs are created by first encoding multi-view images into dense tokens, followed by latent-space K-means clustering and a condensation network to compress these tokens into a smaller representative set. A compute-adaptive transformer renderer then uses a selected number of CLiFTs to synthesize novel views. Extensive experiments on RealEstate10K and DL3DV show that CLiFT achieves competitive rendering quality with fewer tokens or storage than baselines like LVSM, MVSplat, and DepthSplat. The paper highlights the flexibility of CLiFTs in handling different compute budgets and includes ablation studies that validate the importance of its components.

**Questions:**

1. Could the authors provide results comparing CLiFT to both the original (full-capacity) LVSM checkpoints and the decoder-only LVSM variant? This would help quantify the absolute performance improvement of CLiFT over the state-of-the-art and clarify whether CLiFT’s advantages persist under stronger baselines. Including these results would strengthen the evidence for CLiFT’s contribution.

2. Can the authors report rendering FPS or latency metrics for MVSplat and DepthSplat under the same hardware conditions used for CLiFT? This would contextualize CLiFT’s rendering efficiency and its practical deployment value, especially for real-time or resource-constrained applications.

**Ethical Concerns:**

["NO or VERY MINOR ethics concerns only"]

**Final Justification:**

The authors have thoughtfully addressed my concerns, and the newly incorporated information enhances the clarity and understanding of the proposed method. In light of its potential to inspire further research in this area, I am pleased to raise my rating to accept.

**Limitations:**

Yes

**Paper Formatting Concerns:**

In line 44, "... construct CLiFTs At test time",  a full stop was missed.

**Quality:**

2

**Strengths And Weaknesses:**

**Strengths**
1. The proposed CLiFT framework introduces a compact, token-based light field representation that enables dynamic control over rendering quality, speed, and memory footprint. This feature is lacking in many prior approaches.

2. Unlike baselines that require retraining for different compute budgets or scene complexities, CLiFT uses a single model that generalizes across different token counts and rendering settings.

3. Experiments on RealEstate10K and DL3DV show that CLiFT achieves similar or better visual quality (e.g., PSNR) with significantly fewer tokens, demonstrating strong compression capability.

4. The inclusion of detailed ablation studies validates the importance of the latent K-means and neural condensation stages, and shows how each component contributes to rendering quality and efficiency.

5. The method is well-documented with a clean transformer-based architecture, and training/inference pipelines are described in sufficient detail to support reproducibility.

**Weaknesses**

1. While the paper retrains LVSM in a smaller, fairer configuration, results from the original high-capacity LVSM checkpoints should also be presented to contextualize the performance gap. Moreover, only the encoder-decoder variant is compared, despite decoder-only LVSM achieving better results in the original work.

2. Rendering speed (FPS) is compared primarily across CLiFT variants, but missing comparisons to feed-forward baselines like MVSplat and DepthSplat weakens the claim of compute-efficiency, especially since those are designed for real-time rendering performance.

3. Despite compression, the total number of tokens (e.g., 4096 for storage and rendering) and the transformer backbone may still be prohibitive for low-latency or mobile deployment. In addition, more input views will also significantly increase the number of tokens under the dense token setting of this paper.

---

> ### Author Rebuttal · Authors · 2025-07-31
>
> We thank the reviewer for the constructive comments and the questions.
>
> **W1&Q1: Compared with LVSM decoder-only and full capacity LVSM checkpoints.**
>
> - **Compared with a decoder-only LVSM.** We trained a decoder-only LVSM model using 12 blocks and the same hyperparameter settings as our method. While it achieves strong rendering metrics due to having more decoder layers, it is extremely slow, running at only 8.7 FPS. In practice, it is **5–15× slower** than our CLiFT renderer and requires **2–5× more GFLOPs**, indicating substantially higher computational cost. As a result, it is unsuitable for real-time rendering on standard GPUs or mobile devices.
>
>
>     |  | PSNR ↑ | SSIM ↑ | LPIPS ↓ | FPS ↑ | GFLOPs ↓ |
>     | --- | --- | --- | --- | --- | --- |
>     | LVSM-D | 30.83 | 0.919 | 0.088 | 8.7 | 194.2 |
>     | Ours | 29.55 | 0.906 | 0.100 | 54.3 | 70.6 |
> - **Compared with the full capacity LVSM.** The official “full capacity” LVSM decoder‑only model has 24 transformer blocks (≈2× our 12‑block model) and is trained with 2 input views. For reference, we run inference on RE10K with 4 input views using this full capacity checkpoint (no retraining). Note that this checkpoint is trained for 2‑view inputs but evaluated with 4 views, so the extra views are not fully exploited. A 24-block decoder‑only LVSM trained and tested with 4 input views would achieve higher performance.
>
>
>     |  | PSNR ↑ | SSIM ↑ | LPIPS ↓ |  |
>     | --- | --- | --- | --- | --- |
>     | LVSM-D (24 blocks) | 30.16 | 0.920 | 0.082 |  |
>
> **W2&Q2: Compared with rendering FPS with MVSPlat and Depthsplat.**
>
> We tested splatting-based methods on the same hardware, and they can achieve 254.9 FPS. In general, they are significantly faster than reconstruction-free methods that use a neural renderer (LVSM and CLiFT), and this is exactly the main advantage of having explicit 3D representations. We will revise the main paper to include these results.
>
> **W3: Prohibitive for low-latency or mobile deployment as the number of input images increases.**
>
> We respectfully disagree that this is a limitation of our method. CLiFT supports adjusting the number of tokens at inference time, enabling better speed–quality trade-offs than other reconstruction-free methods, such as LVSM.
>
> As shown in Table 1 of the main paper, even with more input images, our rendering pipeline can select only the relevant subset of tokens for the target view, thereby significantly reducing computation while maintaining quality. This advantage would be more critical for larger scenes with more input views, as the necessary number of CLiFTs for a given query view does not increase with the scale of the scene or the number of input views.
>
> **Paper Formatting.**
>
> We thank the reviewer for catching the typos, and we will do a careful pass in the revision to fix all these issues.

---

> > ### Comment · Reviewer_6dxz · 2025-08-04
> > **Author response addressed most of my concerns**
> >
> > I appreciate the authors' transparency in presenting additional LVSM results and providing a rendering speed comparison with 3DGS approaches. While there is a minor discrepancy regarding the number of tokens when handling more views, the overall contribution of this work is evident in its early exploration of efficiency improvements through 3D geometry cues, as opposed to relying solely on full cross-attention mechanisms in feed-forward implicit view synthesis.
> >
> > With the supplementary information provided in the rebuttal, I believe this work has the potential to inspire further research in this area. In light of this, I am inclined to raise my rating to accept as a gesture of positive support for the paper.

---

> ### Author Response · Authors · 2025-08-04
>
> Thank you so much for your valuable feedback. We are pleased that our rebuttal addressed your concerns and sincerely appreciate the time and effort you put into reviewing our paper.

---

### Official Review · Reviewer_W9Zr · 2025-06-29

**Clarity:** 3
**Significance:** 3
**Originality:** 2
**Rating:** 4
**Confidence:** 4

**Summary:**

This work improves large view synthesis model (LVSM) by compressing the multi-view tokens with Kmeans clustering. Previous LVSM attends on tokens of all input views or globally encoded tokens, which leads to expensive computation or pool quality due to too much or few condition tokens. This work cluster the pre-view token into a fixed set and design an attention network to refine the clustered tokens. The clustered tokens offer the decoder a smaller but diverse set token to render novel-view. Experimental results show good tradeoff of the speed and quality by varying the cluster size.

**Questions:**

1. Unfair evaluation. The proposed method use more images for training and testing comparing to the other baselines (L181-184). The result with aligned number of training images should be reported either by ablation of the proposed method or by increasing the number of training images of baselines. If being able to train on more views is one merit of the proposed method, the paper should explain and discuss the reason. The set of testing images should be aligned for all methods. It's not reasonable to compare numbers evaluated on different set of images.

2. Paper presentation. In the method section, there is no need to filled in the hyperparameters with specific numbers if the proposed method can only work at that specific setup. Like the image resolution can be other than 256x256 so it's better to just state HxW. The figure of the Neural Condensation part in Fig.1 is hard to understand. One good figure example is VGGT to illustrate the operation of intra and inter group attention.

**Ethical Concerns:**

["NO or VERY MINOR ethics concerns only"]

**Final Justification:**

My main concern about the evaluation is addressed. The discussion and the new results in the other responses are really insightful. Overall, it's a technical solid paper, the novelty and improvement is sufficient so I increase my rating from negative to positive.

**Limitations:**

One limitation missing discussion is that the fixed number of clusters may not have good speed-quality tradeoff for different scenes. An adaptive strategy can be developed to automatically or dynamically determine the number of cluster based on the training view observations.

**Paper Formatting Concerns:**

The paper format looks right.

**Quality:**

2

**Strengths And Weaknesses:**

The proposed method serves as a simple but effective efficiency improvement on the recent interesting topic: large novel-view synthesis model with minimum 3d inductive bias. The proposed clustering strategy to aggregate multi-view tokens is a reasonable extension, which is expected to give better speed-quality tradeoff and the experimental results by this work fit this expectation. Overall, I view this work as a good increment on existing trend.

On the other hand, there is also nothing very special that prompts me to give high rating to this work. Given the concerns I have in the Questions sections, I currently leans to give negative rating.

---

> ### Author Rebuttal · Authors · 2025-07-31
>
> We thank the reviewer for the detailed feedback.
>
> **Q1: Unfair Evaluation.**
>
> We are sorry for the confusion. We use the same number of input views as the baselines when evaluating performance. Details of the comparison setup are provided in the baseline section (L 201–205). L181-184 describes the input setting of the original papers and leads to confusion. We will fix this in the revision.
>
> - **RealEstate10K:** For LVSM, we retrained their model from scratch using the same 4‑view input setting as ours (L 207). For MVSplat and DepthSplat, which predict Gaussian splats per pixel, the storage size grows with the number of input views and resolution. While we also provide experiments with the same input view settings on DL3DV for direct comparison, here on RE10K we use their official 2‑view checkpoints to examine **PSNR at the same storage size**, placing their storage in a comparable range.
> - **DL3DV**: Our 4-view and 6-view models are evaluated directly against the publicly available 4-view and 6-view checkpoints of MVSplat and DepthSplat (L211), ensuring fair evaluation settings.
>
> **Q2: Paper presentation.**
>
> The footnote on page 4 explains the resolution settings. In the revision, we will use explicit hyperparameter notations (e.g., H×W) for clarity. The non-centroid tokens do not participate in the inter-cluster self-attention in our design, so the VGGT-style visualization doesn’t apply directly to our case. We will improve the clarity in the revision.
>
> **Limitations: Can not automatically select the number of clusters for different scenes.**
>
> We thank the reviewer for raising this interesting discussion point. Our design is motivated by scenarios with predefined compute or storage constraints, such as a target FPS or GPU memory limit. Nonetheless, our system can be readily extended to automatically select the number of clusters (i.e., tokens). One simple approach is to encode the scene using multiple values of K and select the one that satisfies a desired validation PSNR threshold. A more elegant solution—such as learning to predict K during the forward pass based on a target PSNR—remains an interesting direction for future work.

---

> > ### Comment · Reviewer_W9Zr · 2025-08-04
> >
> > Thanks for the response, which addresses my main concern about evaluation.
> >
> > The maximum storage usage of baseline in Fig.2 is less than 45MB, so I think comparing PSNR at the same storage size is not necessary as the storage size doesn't seem to be the main concern of a method in current GPU.
> >
> > My main concern is addressed and the discussion and the new results in the other responses are really insightful. I will raise my rating.

---

> > > ### Author Response · Authors · 2025-08-04
> > >
> > > We sincerely appreciate the time and effort you dedicated to reviewing our paper. While the scene representation size has little impact when the number of views is small, it can become a noticeable overhead as the number of views increases (e.g., in larger scenes) or when many scenes need to be stored for potential rendering applications. We will revise the paper to include further discussion on this point.

---

### Official Review · Reviewer_g7Ys · 2025-06-29

**Clarity:** 3
**Significance:** 2
**Originality:** 3
**Rating:** 4
**Confidence:** 4

**Summary:**

The authors present CLiFTs (Compressed Light-Field Tokens), a novel and data-efficient neural scene representation that addresses the high memory demands of existing neural rendering methods. Built on a Transformer encoder-decoder architecture, the framework introduces an encoding stage that constructs CLiFTs by applying K-means clustering in latent space to identify feature centroids from multi-view inputs. A proposed condensation network is then used to compress neighboring features around each centroid.

This approach enables significantly reduced memory storage compared to traditional feed-forward neural rendering pipelines, while still producing high-quality rendered images. The method is evaluated on two benchmark datasets, RealEstate10K and DL3DV, against three representative baselines, LVSM, MVSplat, and DepthSplat. CLiFTs demonstrate a good trade-off between rendering quality and data efficiency, particularly outperforming prior methods from the perspective of storage and computational cost.

**Questions:**

- It would be helpful to include an experiment that evaluates how the method performs across scenes with varying complexity. Since both datasets used in the paper may contain scenes of similar visual complexity, it’s hard to tell how well CLiFT generalizes to more diverse scenarios, and its range of scenes' complexity that CLiFT can cover.

- Including an ablation study that replaces the K-means clustering with simple alternative, such as grouping locally in patch style, and uses only the neural condensation component would be valuable to understand the proposed methods' contribution.

- The paper could benefit from more explanation around the architecture and design choices of the condensation network. A brief motivation for the layer/module design and why certain decisions were made would help readers better understand how this component contributes to the overall method.

- Does the model maintain good performance when more input views are provided? It would be good to include an experiment that tests the model’s generalization ability with a larger number of input views, especially if the number of CLiFT tokens is reduced at the same time. Showing the upper-bound of models' performance within the trade-off of the number of input views and tokens could further highlight its strength as a data-efficient approach.

Based on the authors' rebuttal, I'm willing to revise my initial rating.

**Ethical Concerns:**

["NO or VERY MINOR ethics concerns only"]

**Final Justification:**

The authors have adequately addressed most of the concerns I raised in my initial review, providing appropriate results and detailed explanations in the rebuttal. I believe the paper is of a quality that can receive a rating close to acceptance.

**Limitations:**

Yes.

**Paper Formatting Concerns:**

I found no major formatting issues in the paper.

**Quality:**

2

**Strengths And Weaknesses:**

> Strengths
- The paper is clearly written and easy to read. The target problem is well-motivated, and the evaluation metrics are appropriately chosen to show the effectiveness of the proposed method.
- The proposed method shows strong performance in terms of data efficiency, outperforming existing baselines both quantitatively and qualitatively.

> Weaknesses
- The rationale for choosing a light field representation is not sufficiently discussed. It's unclear what specific advantages it brings over alternative representations, especially in comparison to the baselines.
- There is limited discussion on potential issues with using K-means clustering in latent space. It would be helpful to include analysis or experiments addressing whether the method generalizes well across diverse scene types, e.g., across the scenes of varying complexity.
- The condensation network is introduced as a core component, but details on its architecture and design choices are insufficient. More justification or ablation results would strengthen the quality of paper.

---

> ### Author Rebuttal · Authors · 2025-07-31
>
> We thank the reviewer for the constructive and detailed feedback.
>
> **W1: Advantages of light field representation.**
>
> **Compared with Reconstruction-Based Methods (e.g., DepthSplat [34]):**
>
> A key advantage of our *light field representation* is that it is *reconstruction-free*—it does not require hand-engineered intermediate geometric or photometric representations such as 3D Gaussians or spherical harmonics (L30–32). Instead, the entire pipeline operates on learned “tokens,” making the representation fully data-driven and flexible. This aligns with an emerging trend in novel view synthesis (NVS), as exemplified by methods like LVSM [12].
>
> This token-based formulation offers two main advantages:
>
> 1. It provides a straightforward path to extend NVS capabilities by integrating with other token-based networks, such as LLMs or VLMs, enabling applications like text-controlled rendering (e.g., adjusting style or lighting via prompts) or embodied AI (e.g., using NVS tokens as pretraining input for policy learning in agents, as done in VLA).
> 2. It improves the system’s ability to handle complex scenes—both geometrically (e.g., thin structures) and photometrically (e.g., view-dependent effects and potentially dynamic content)—which are typically challenging for methods relying on explicit reconstruction.
>
> We thank the reviewer for raising this point and will revise the paper to better highlight these advantages of the light field representation (L30–32).
>
> **Compared with LVSM:**
>
> Like our approach, LVSM is also reconstruction-free and operates entirely on token-based representations. The key distinction is in how the scene is represented. LVSM models the scene using all tokens from all input images, without explicit spatial organization. In contrast, our *light field tokens* (CLiFTs) are explicitly associated with rays, encoding localized appearance and geometry. This ray-based, localized design is fundamental to our contributions: it allows the scene to be represented as a variable-length list of compressed tokens and enables *compute-adaptive rendering* by selecting only the subset of tokens relevant to a target view. We will revise the paper to clarify these differences more explicitly (L91–99).
>
> **W2&Q2: Potential issues with K-means and the ablation study for its effectiveness**
>
> We thank the reviewer for raising this important question. We agree that the latent K-means clustering may not be optimal, as it is based on heuristics and not explicitly optimized for the final objective—rendering quality. To assess its effectiveness, we compared latent K-means with a simple patch-wise clustering baseline that uses non-overlapping local patches. The table below reports PSNR scores for both methods, with and without neural condensation, under two token budget settings (256 and 1024 tokens):
>
> |  | 256 | 1024 |
> | --- | --- | --- |
> | Patch-wise | 22.64 | 27.48 |
> | Patch-wise with condenser | 22.97 (+0.33) | 27.55 (+0.07) |
> | K-means | 24.46 | 28.17 |
> | K-means with condenser | 25.21 (+0.75) | 28.41 (+0.24) |
>
> These results show that latent K-means consistently outperforms patch-wise clustering at both budget levels, with particularly notable improvements under stronger compression (e.g., +2.24 PSNR at 256 tokens with condensation). This advantage arises because patch-wise clustering allocates capacity uniformly across regions, while latent K-means adaptively places tokens in more informative areas, thereby better preserving quality when token budgets are tight. We will include this ablation study in the final version of the paper.
>
> **W3&Q3: Motivation, design choice, and ablation of the condensation network.**
>
> We thank the reviewer for their feedback. The goal of the neural condensation module is to extract information from *to-be-discarded tokens and* condense it into a smaller set of representative tokens. The architecture is intentionally designed to achieve this: self-attention facilitates information exchange among cluster centroids, while cross-attention allows each centroid to absorb relevant information from the tokens assigned to it. We will clarify this explanation in Lines 139–150 of the revised paper.
>
> Our ablation study in Figure 3 supports this design choice. When the condenser module is removed, model performance degrades significantly under high compression rates. This is expected, as the majority of token information is lost without condensation, highlighting the importance of this module for effective compression.
>
> **Q1: Method performs across scenes with varying complexity**
>
> We followed prior works in selecting the RealEstate10K and DL3DV datasets, which offer varying levels of complexity. RealEstate10K consists of indoor scenes with relatively smooth camera trajectories, whereas DL3DV presents more challenging conditions, including outdoor environments and complex camera motion.
>
> Our model demonstrates strong performance on both datasets, highlighting its ability to generalize across diverse scenarios. A visual comparison of the rendering results on these datasets is included in the supplementary video (demo.mp4).
>
> **Q4: The performance of more input views.**
>
> We conducted additional ablation studies by increasing the number of input views without retraining for that setting, and observed little to no performance gain—consistent across LVSM, DepthSplat, and our method.
>
> *RE10K (ours & LVSM, trained for a fixed 4 view count):*
>
> |  | 4 views | 6 views | 8 views |
> | --- | --- | --- | --- |
> | LVSM | 28.95 | 29.16 | 29.19 |
> | CLiFT | 29.40 | 29.48 | 28.92 |
>
> *DL3DV (DepthSplat pretrained with 2~6 views):*
>
> |  | 6 views | 8 views | 10 views |
> | --- | --- | --- | --- |
> | DepthSplat | 25.43 | 25.16 | 24.86 |
>
> In addition, the recent paper PRoPE [A] also shows that the current camera encoding approach struggles to generalize to more input views. The proposed relative camera encoding could help address this problem.
>
> ---
>
> [A] Li, R., Yi, B., Liu, J., Gao, H., Ma, Y., & Kanazawa, A. (2025). Cameras as Relative Positional Encoding. arXiv preprint arXiv:2507.10496.

---

> > ### Comment · Reviewer_g7Ys · 2025-08-06
> >
> > Thank you to the authors for their thoughtful and detailed rebuttal. I appreciate the clarifications provided, particularly regarding the K-means discussion and the role of the condensation network. I hope these points will be clearly reflected in the revision.
> >
> > Most of my concerns are adequately addressed, and I will raise my initial score.

---

> > > ### Author Response · Authors · 2025-08-06
> > >
> > > We are delighted that our rebuttal addressed your concerns. We thank you for your valuable feedback. We will include the new ablation and K-means discussion in the revision.

---

### Official Review · Reviewer_gUfU · 2025-07-02

**Clarity:** 3
**Significance:** 3
**Originality:** 4
**Rating:** 5
**Confidence:** 2

**Summary:**

This paper introduces CLiFT, a novel approach for neural rendering that represents a scene using a set of "compressive light-field tokens." The primary contribution is a framework that is both data-efficient and adaptive, allowing a single trained model to trade-off between storage size, rendering quality, and computational speed. The method consists of a three-step construction process: 1) a multi-view Transformer encoder converts input images and poses into initial "Light Field Tokens" (LiFTs); 2) a latent-space K-means algorithm clusters these tokens to select a compact set of representative "centroid" rays, effectively reducing geometric and appearance redundancy ; and 3) a "neural condenser" further compresses information from all tokens in a cluster into these centroids, creating the final CLiFTs. At inference time, a novel view is synthesized by a Transformer decoder that uses a subset of these CLiFTs, selected based on proximity to the target view's rays. Experiments on the RealEstate 10K and DL3DV datasets show that CLiFT achieves comparable or better rendering quality than state-of-the-art methods like LVSM and DepthSplat, but with significantly smaller data sizes.

**Questions:**

Regarding the Two-Stage Training: The paper describes a two-stage training process where the encoder is trained first and then frozen.  What is the motivation behind this design instead of end-to-end training?

On Failure Cases: The limitations section mentions that the method struggles with camera poses that are very different from the training data.  Is there any mechanism within the model to estimate the uncertainty or expected quality of a rendering for a given target pose? This would be very useful for practical applications.

**Ethical Concerns:**

["NO or VERY MINOR ethics concerns only"]

**Final Justification:**

The author addressed my concerns by providing the reference and justification, as well as discussing the generalization limitations and multi-stage training complexity. I believe the author is aware of these limitations and has a grasp on addressing them. Therefore, I uphold my initial decision to Accept.

**Limitations:**

Yes, the authors discuss limitations in Section 5, pointing out failure cases related to out-of-distribution camera motions and insufficient view coverage in the input images.  This discussion is fair and provides a clear picture of the method's boundaries.

**Paper Formatting Concerns:**

I have not seen any so far.

**Quality:**

3

**Strengths And Weaknesses:**

- Strengths:
1. Adaptive Representation: The most compelling strength of CLiFT is its ability to operate across a spectrum of data sizes and compute budgets with a single trained model. Users can dynamically adjust the number of tokens for storage or rendering to balance quality and speed. This is a significant practical advantage over existing methods, which typically require retraining a separate model for each desired data size or quality level.

2. High Compression Efficiency: The paper demonstrates impressive data compression. CLiFT achieves comparable PSNR to MVSplat and DepthSplat with 5-7x less data and to LVSM with 1.8x less data.  This efficiency is achieved through the intelligent reduction of redundant information, making the representation highly compact.

3. Principled Token Selection and Condensation: The use of latent-space K-means to identify and select representative rays is a more principled approach to compression than uniform or random sampling. The ablation studies in Figure 3 clearly show that this selection strategy is critical for maintaining quality at high compression rates. The subsequent neural condensation step further refines these tokens, leading to consistent performance gains.

- Weakness:
1. Generalization Limitations: As acknowledged by the authors, the method struggles when faced with camera motions that deviate significantly from the training distribution or when target views are not well-covered by the input images.  This highlights a potential weakness in generalization, which is a common challenge for learning-based novel view synthesis methods.

2. Multi-Stage Training Complexity: The training pipeline is quite complex, requiring a two-stage process where the multi-view encoder is trained first, then frozen, followed by a second stage to train the condenser using pre-computed K-means assignments.  This separation may introduce optimization challenges and could be a barrier to reproducibility and future extensions compared to a single, end-to-end trainable system.

---

> ### Author Rebuttal · Authors · 2025-07-31
>
> Thank you for the constructive comments and questions.
>
> **W1: Generalization limitations on camera poses.**
>
> We agree that generalization to novel camera poses is a key challenge for learning-based novel view synthesis (NVS), particularly for reconstruction-free approaches that do not rely on hand-crafted geometric or photometric representations. Figure 6 shows our typical failure cases. We observe that this generalization issue often occurs for RE10K, which has limited camera motion variations and results in poor generalization (L267-270). DL3DV has more complex camera trajectories, and we observe far fewer such failures, indicating that the limitation is mainly due to training data coverage. We will clarify the point in the paper.
>
> **W2&Q1: Multi-Stage training complexity.**
>
> We acknowledge that single-stage training presents challenges, as latent K-means is not readily differentiable. This naturally leads to a decoupled training process, where the encoder is trained separately from the condensation and rendering modules. While end-to-end training using a form of “soft clustering” is an exciting direction for future work, we emphasize that multi-stage training is a practical and effective strategy, often leading to more stable optimization and easier modular development. For example, latent diffusion models typically train the autoencoder and the denoising network in two separate stages. Importantly, at inference time, our method only requires running the encoding (latent K-means, neural condensation) for each scene once, and the rendering module is very fast.
>
> **Q2:  Mechanism to estimate the quality of rendering given a target pose**
>
> We thank the reviewer for the valuable suggestions. As in many novel view synthesis works, rendering quality is typically higher when the target camera pose is closer to the input camera pose and degrades as the distance increases. A simple approach is to compute the translation and rotation differences between the target pose and the closest input pose, which can serve as a rough estimate of rendering quality. We will include related results in the paper.

---

> > ### Comment · Reviewer_gUfU · 2025-08-07
> >
> > The authors have effectively addressed my concerns. They explained the generalization limitations by referencing typical failure cases shown in Figure 6. They also justified the multi-stage training complexity and offered an insightful suggestion of a heuristic to estimate rendering quality based on pose differences.
> > Given these clarifications, I believe the paper is well positioned for further improvement and I maintain my decision to Accept.

---

> > > ### Author Response · Authors · 2025-08-07
> > >
> > > Thank you for your feedback and support of our work! We are glad our responses addressed your concerns.

---

### Decision · Program_Chairs · 2025-09-17

**Decision:**

Accept (spotlight)

**Comment:**

This paper received uniformly positive reviews. It presents an interesting idea that builds upon an LVSM-like architecture by introducing compressive tokens, thereby enabling computationally efficient rendering. The method demonstrates promising trade-offs between rendering quality and efficiency, which the reviewers found compelling.

Explicit 3D representations, such as 3DGS, enable fast rendering but often with somewhat lower visual quality, whereas LVSM-like models achieve superior rendering quality at the cost of high computational expense. This paper is well positioned in this context, as it proposes a direction that narrows the gap by retaining much of the rendering quality while improving efficiency.

Overall, this paper makes a valuable contribution by advancing an important line of research and opening new opportunities for both explicit and implicit 3D representations. I support its acceptance.